Locally developed models improve the accuracy of remotely assessed metrics as a rapid tool to classify sandy beach morphodynamics

Checon Helio Herminio hchecon@yahoo.com.br 1 2
Shah Esmaeili Yasmina 1
Corte Guilherme N. 2 3
Malinconico Nicole 2
Turra Alexander 2
1 Departament of Animal Biology, Universidade Estadual de Campinas , Campinas , São Paulo , Brazil
2 Oceanographic Institute, Universidade de São Paulo , São Paulo , São Paulo , Brazil
3 Escola do Mar, Ciência e Tecnologia, Universidade do Vale do Itajaí , Itajaí , Santa Catarina , Brazil
Foody Giles
Electronic publication date: 2022 May 17
Publication date: 2022
Volume: 10
Electronic Location ID: e13413
Received 2022 Feb 4; Accepted 2022 Apr 19
Copyright: ©2022 Checon et al.
Copyright year: 2022
Copyright holder: Checon et al.
License: This is an open access article distributed under the terms of the Creative Commons Attribution License, which permits unrestricted use, distribution, reproduction and adaptation in any medium and for any purpose provided that it is properly attributed. For attribution, the original author(s), title, publication source (PeerJ) and either DOI or URL of the article must be cited.
License URL: https://creativecommons.org/licenses/by/4.0/

Keywords: Conditional tree inference, Environmental modelling, Brazilian coast, Coastal management

Funding: The Fundação de Amparo à Pesquisa do Estado de São Paulo No. 2018/22036-0 No. 2017/17071-9 No. 2018/0955-5 FAPESP BIOTA project program No. 2018/19776-2 The FAPESP global climate change research program thematic project No. 2015/03804-9 The Coordenação de Aperfeiçoamento de Pessoal de Nível Superior (CAPES) The Conselho Nacional de Desenvolvimento Científico e Tecnológico (CNPq) No. 309697/2015-8 No. 310553/2019-9 No. 165320/2020-6 Fundacao Grupo Boticario No. 1133_20182 This study was funded by the Fundação de Amparo à Pesquisa do Estado de São Paulo through scholarship grants to Helio Herminio Checon (No. 2018/22036-0), Guilherme N Corte (No. 2017/17071-9), Yasmina Shah Esmaeili (No. 2018/0955-5) and through the funding linked to the FAPESP BIOTA project program (No. 2018/19776-2) and the FAPESP global climate change research program thematic project (No. 2015/03804-9); and the Coordenação de Aperfeiçoamento de Pessoal de Nível Superior (CAPES) through scholarship grants to Nicole Malinconico and the Conselho Nacional de Desenvolvimento Científico e Tecnológico (CNPq) for the scholarship grants to Alexander Turra (No. 309697/2015-8 and No. 310553/2019-9) and Guilherme N Corte (No. 165320/2020-6). This project was also funded by Fundacao Grupo Boticario (No. 1133_20182). The funders had no role in study design, data collection and analysis, decision to publish, or preparation of the manuscript.

==============================
Classification of beaches into morphodynamic states is a common approach in sandy beach studies, due to the influence of natural variables in ecological patterns and processes. The use of remote sensing for identifying beach type and monitoring changes has been commonly applied through multiple methods, which often involve expensive equipment and software processing of images. A previous study on the South African Coast developed a method to classify beaches using conditional tree inferences, based on beach morphological features estimated from public available satellite images, without the need for remote sensing processing, which allowed for a large-scale characterization. However, since the validation of this method has not been tested in other regions, its potential uses as a trans-scalar tool or dependence from local calibrations has not been evaluated. Here, we tested the validity of this method using a 200-km stretch of the Brazilian coast, encompassing a wide gradient of morphodynamic conditions. We also compared this locally derived model with the results that would be generated using the cut-off values established in the previous study. To this end, 87 beach sites were remotely assessed using an accessible software (i.e., Google Earth) and sampled for an in-situ environmental characterization and beach type classification. These sites were used to derive the predictive model of beach morphodynamics from the remotely assessed metrics, using conditional inference trees. An additional 77 beach sites, with a previously known morphodynamic type, were also remotely evaluated to test the model accuracy. Intertidal width and exposure degree were the only variables selected in the model to classify beach type, with an accuracy higher than 90% through different metrics of model validation. The only limitation was the inability in separating beach types in the reflective end of the morphodynamic continuum. Our results corroborated the usefulness of this method, highlighting the importance of a locally developed model, which substantially increased the accuracy. Although the use of more sophisticated remote sensing approaches should be preferred to assess coastal dynamics or detailed morphodynamic features (e.g., nearshore bars), the method used here provides an accessible and accurate approach to classify beach into major states at large spatial scales. As beach type can be used as a surrogate for biodiversity, environmental sensitivity and touristic preferences, the method may aid management in the identification of priority areas for conservation.

Introduction

Sandy beaches occupy a third of the world’s non-frozen coastline (Luijendijk et al., 2018) and are the most used coastal ecosystem by human populations (McLachlan, Defeo & Short, 2018). The increasing urbanization in coastal areas, however, exerts great pressure on these ecosystems, which are trapped between potential impacts from both the terrestrial and marine environments (Schlacher et al., 2008; Nel et al., 2014). Currently, nearly a quarter of the world’s sandy beaches are eroding at rates higher than 0.5 m/yr, including the majority of sandy shores in marine protected areas (Luijendijk et al., 2018). Given that sandy beaches provide ecosystem services that are essential for human populations (Schlacher et al., 2008; Barbier, 2011; Barbier, 2017), worldwide efforts are necessary to monitor and preserve these ecosystems.

Over the past decades, researchers have shown that sandy beach functioning is strongly linked to local morphodynamic characteristics (Lercari, Bergamino & Defeo, 2010; Bergamino et al., 2013). The interaction between geological (e.g., sediment, beach slope) and hydrodynamic features (e.g., wave height, tide range) determines a morphodynamic continuum of beach types, ranging from reflective (i.e., coarse-grained beaches, steep profiles, no surf-zone) to dissipative beaches (i.e., fine-grained beaches, flat slopes, extensive surf-zone), across multiple intermediate states (Wright & Short, 1984). To allow the distinguishing of morphodynamic units across multiple scales, beaches are classified into these morphodynamic types (Short & Jackson, 2013), one of the most important features in predicting ecological processes and biodiversity patterns in beaches (McLachlan & Dorvlo, 2005; McLachlan & Defeo, 2017). Abundance and diversity of macrobenthic species tend to increase from reflective to dissipative beaches, with shifts in dominance of groups and composition of assemblages (Barboza et al., 2012; Defeo & McLachlan, 2013; Checon et al., 2018). The morphodynamic type is therefore likely to influence the presence of biodiversity-related ecosystem services such as food supply, genetic diversity, biomass stock, and nutrient cycling (McLachlan, Defeo & Short, 2018). Beach type may also influence the potential of beaches to serve as nursery grounds for fishes, which are suggested to be more common in dissipative beaches (Oliveira & Pessanha, 2014; Shah Esmaeili et al., 2021), and nesting grounds for transitory and resident fauna, such as the preference of turtles for intermediate beaches (Siqueira et al., 2021).

Identifying beach morphodynamics may also be important to support management strategies (Jimenez et al., 2007; McLachlan et al., 2013). Morpho and hydrodynamics characteristics that vary among beach types, such as beach width and wave height, affect the potential of beaches for tourism and recreational activities, a highly relevant economic asset for countries worldwide (Philips & House, 2009; Onofri & Nunes, 2013). Beach type is also related to differential susceptibility to impacts from anthropic and natural disturbances (Harris et al., 2015; Santos & Turra, 2017; McLachlan, Defeo & Short, 2018). For instance, reflective beaches may be more susceptible to contamination of groundwater by pollutants due to the rapid percolation in coarse sands (Bernabeu et al., 2006), whereas dissipative beaches are more prone to accumulation of marine litter (Tsukada et al., 2021). Beach type also is suggested to affect the effectiveness of bioindicators, an approach commonly used for monitoring beaches (Costa et al., 2022). Thus, a remote classification of beach type can serve as a proxy of the ecological processes, patterns, and services in sandy beaches, as well as in the support of management strategies.

Given the importance of characterizing the morphodynamics of beach ecosystems, multiple indices have been developed to act as surrogates of the beach types, whether based on hydrodynamic features, such as the Dean Parameter (Ω) and Relative Tide Range (RTR) or morphological variables, such as the Beach Index (BI) and the Beach Deposit Index (BDI) (McLachlan & Dorvlo, 2005; Short & Jackson, 2013). Assessment of the variables required to classify beaches were historically made by local sampling, such as the traditional beach profiling method (Emery, 1961) and evaluation of sediment and wave conditions (Schlacher et al., 2008; Short & Jackson, 2013). These surveys, however, usually demand traveling large distances and the collection of a high number of samples, therefore increasing personnel and financial requirements. The development of technologies for remote assessments has helped to reduce these needs, allowing rapid assessments of the conditions and changes on sandy beach ecosystems over large spatial scales in a synoptic way (Kroon et al., 2007; Mars & Houseknecht, 2007; Luijendijk et al., 2018).

Several remote tools have been used, through different techniques using active (e.g., LiDAR, Terrestrial Laser Scanning) and passive sensors (e.g., Argus and hyper-spectral imagery), to characterize the morphodynamic type of beaches (Deronde et al., 2008; Ellenson et al., 2020; Jackson & Short, 2020). These methods often involve expensive equipment and/or software processing of images and spectral analysis, which can provide a detailed identification of features relevant to identify beach types (Splinter, Harley & Turner, 2018; Ellenson et al., 2020), but require a knowledge to or the use of services for treating and processing satellite or aerial images. To provide a more accessible method to classify beaches into major states, Harris, Nel & Schoeman (2011) suggested a method based on the remote measurements of beach features using freely available satellite images from Google Earth. Using conditional tree inferences, they achieved a 93% prediction accuracy of beach morphodynamic type on the South African Coast, proving to be an accessible method to map beach types at large-scales with a good accuracy. Although not providing the same level of detail than methods based on images processing, it allowed for the classification of beaches into major states, a scale that is often used and relevant for ecological studies and management strategies (McLachlan, Defeo & Short, 2018). However, as pointed out by the authors, this method may depend on local geomorphological and oceanographic conditions of biogeographical regions, and its extrapolation pended validation in other areas (Harris, Nel & Schoeman, 2011). Thus far, the method has been used to remotely measure beach morphodynamics in other coastal areas in South Africa (Pattrick & Strydom, 2014), United States (Shanks, Walser & Shanks, 2014), and Australia (Borland et al., 2017), but no effort has been made to test the validity of the initial classification scheme or the need for calibration in different areas.

The objective of this study was to test the effectiveness of remote assessment of beach characteristics in predicting the in situ morphodynamic states. To achieve this goal, we first replicated the method developed by Harris, Nel & Schoeman (2011) on a set of 87 sandy beach sites in Southeast Brazil to test its consistency and the possible need for calibration. This comparison is important to understand whether cut-off values established in Harris, Nel & Schoeman (2011) can be easily applied in different shores, and be used in a global or transregional scale, or whether a local calibration is necessary. After the method validation, we used satellite imagery to classify the morphodynamic type of beaches across the whole North Coast of São Paulo (∼200 km coastline). The result of this study provides important information for local managers and researchers, in addition to indicate the usefulness or limitations of the method outside of its original area and promote its application in sandy shores studies worldwide.

Materials & Methods

Study area

This study was carried out on beaches located on the northern coast of the State of São Paulo, Brazil. This area was chosen due to the heterogeneity of shore morphological features, with a more straight, exposed shore on the southern part, and a sinuous, more sheltered coastline towards the northern part of the study area (Fig. 1). This characteristic was a desirable feature, as it allowed us to evaluate the efficacy of the method on a large range of morphodynamic types, from dissipative to reflective beaches, also including exposed and sheltered shores. Additionally, this area involves a complex environmental and governance setting, as much of the coastal region is located within protected areas, but suffers from threats from urbanization and intense tourism (Pierri Daunt et al., 2021). With approximately 200 beaches located along ∼200 km of coastline, many of these being remote and having a difficult access, methods that can provide a rapid assessment of physical and biological characteristics of beaches, as well as providing baselines for their monitoring, are valuable tools for the management of these ecosystems.

Figure 1 Map of the study area.

indicating the beaches sampled for environmental characterization to develop the classification scheme (train dataset) and beaches used to validate the classification (test dataset).

A total of 29 beaches were sampled to identify the environmental characteristics (i.e., in-situ morphodynamic and hydrodynamic characteristics) between February and June/2019. We divided each beach into three subareas (the middle region and both beach corners) with a minimum distance of 100 m among them, which were further treated as individual sites (n = 87). This division was done due to the great alongshore variation in natural features (i.e., waves, beach slope) which affect beach type. Additionally, human activities in beaches may be directed towards specific areas, especially in extensive beaches (Alexandrakis, Manasakis & Kampanis, 2015; Machado et al., 2016), and management may benefit from considering the local variability within beaches.

Assessment of beach physical characteristics

To evaluate beach morphodynamics in situ, beach profile and slope (inclination of the intertidal), and granulometric parameters were assessed at each sampling site. Beach profile was measured from the supratidal to the end of the swash zone following the method described by Emery (1961), during low spring tides. The width of the intertidal area was registered in the profiling and the slope, calculated using the trigonometric relationships, estimated as 1/tg(β). To estimate the mean grain size, five sediment samples were collected along the beach profile. In the laboratory, after drying, they were sieved into twelve granulometric fractions and the mean grain calculated based on Folk & Ward (1957). Sites were classified into five morphodynamic types: dissipative, intermediate-dissipative, intermediate, intermediate-reflective, and reflective. This characterization was based on the environmental features measured at each site and further supported by the calculation of the Beach Index (BI, McLachlan & Dorvlo, 2005).

Remote assessment of beach characteristics

Remote assessment of beach conditions was done by measuring specific features on satellite images. Images were obtained from the Google Earth Software (CNES/Airbus, Maxar Technologies), as it is the most user-friendly and accessible source, whose application can be standardized across regions, facilitating a rapid and easily applicable assessment by a wider range of users. The date of the available images was chosen as close as possible to the date of sampling of each beach.

Similar to Harris, Nel & Schoeman (2011), the following characteristics were assessed: beach intertidal width, surf-zone width, number of waves and bores in the surf-zone, and degree of exposure (Table 1, Fig. 2). All these metrics are expected to change within the beach morphodynamic continuum and are thus considered good proxies for classifying beach type. Beach intertidal width was measured as the distance from the drift line to the lowest position of the swash. Surf-zone width was measured as the distance from the innermost to the outermost swash edge. The number of waves and bores within the surf-zone was determined by distinguishing the number of visible ripples. Surf-zone type, a variable included in Harris, Nel & Schoeman (2011), was initially considered, but we found this variable to be difficult to identify in many instances, especially in standardizing across a set of temporal images, and thus we excluded this variable from the analysis. Finally, the degree of exposure was coded from 0 (very sheltered) to 4 (very exposed) (See Fig. S1 for image distinctions among types). The degree of exposure was established by observing the shape and degree of curvature of the headlands, as well as the presence of nearshore islands or other formations that could affect wave exposure (Turnbull et al., 2018).

Table 1 Environmental characterization obtained for the 87 sites across 29 beaches.

Mean ± SD values are given for each beach, based on the average data among sites. Please, see Table S1 for the values obtained for each site Location refers to the municipality where the beach is located (SS: São Sebastião, CA: Caraguatatuba, UB: Ubatuba). Beach type is given for each sector and is coded as D: Dissipative, ID: Intermediate Dissipative, I: Intermediate, IR: Intermediate Reflective, R: Reflective.

 	 	 	 	 	 	Beach type	
Beach	Location	Intertidal width (m)	BI	Slope (s)	Mean grain size (ϕ)	Site 1	Site 2	Site 3	
Baleia	SS	72 ± 10.5	2.47 ± 0.08	0.02 ± 0.01	2.98 ± 0.09	D	D	D	
Barequeçaba	SS	87 ± 7.7	2.53 ± 0.02	0.02 ± 0.01	3.34 ± 0.01	D	D	D	
Barra Seca	UB	17 ± 2.9	1.93 ± 0.02	0.06 ± 0.01	2.78 ± 0.23	I	I	I	
Boiçucanga	SS	18 ± 2.8	1.21 ± 0.11	0.16 ± 0.01	1.36 ± 0.43	R	R	R	
Boraceia	SS	85 ± 8.6	2.58 ± 0.05	0.01 ± 0.01	3.05 ± 0.21	D	D	D	
Caçandoca	UB	25 ± 0.0	1.67 ± 0.05	0.08 ± 0.02	1.81 ± 0.28	IR	IR	IR	
Capricórnio	CA	20 ± 0.0	1.22 ± 0.05	0.17 ± 0.04	1.39 ± 0.12	R	R	R	
Cidade	CA	28 ± 10.4	1.82 ± 0.28	0.08 ± 0.04	2.40 ± 0.06	I	ID	D	
Domingas Dias	UB	27 ± 2.9	1.75 ± 0.03	0.08 ± 0.01	2.34 ± 0.31	I	I	I	
Dura	UB	85 ± 8.7	2.57 ± 0.07	0.02 ± 0.01	3.16 ± 0.28	D	D	D	
Fazenda	UB	62 ± 5.8	2.38 ± 0.04	0.02 ± 0.01	3.02 ± 0.14	D	D	D	
Félix	UB	22 ± 5.8	1.63 ± 0.05	0.08 ± 0.03	1.79 ± 0.55	R	IR	I	
Fortaleza	UB	28 ± 2.9	2.05 ± 0.02	0.05 ± 0.01	2.64 ± 0.21	ID	ID	ID	
Grande	UB	45 ± 5.0	1.98 ± 0.05	0.05 ± 0.01	2.28 ± 0.12	ID	ID	ID	
Guaecá	SS	35 ± 5.0	1.96 ± 0.09	0.05 ± 0.01	2.25 ± 0.21	ID	ID	ID	
Itaguá	UB	20 ± 5.0	1.71 ± 0.37	0.10 ± 0.07	2.23 ± 0.61	IR	I	ID	
Itamambuca	UB	47 ± 2.9	1.93 ± 0.28	0.06 ± 0.02	2.25 ± 0.34	ID	ID	ID	
Jureia	SS	22 ± 2.9	1.34 ± 0.01	0.15 ± 0.01	1.65 ± 0.10	R	R	R	
Lagoinha	UB	38 ± 21.0	2.07 ± 0.36	0.05 ± 0.04	2.61 ± 0.16	D	ID	I	
Perequê-Mirim	UB	25 ± 5.0	2.05 ± 0.22	0.05 ± 0.02	2.74 ± 0.07	ID	ID	ID	
Porta	CA	23 ± 2.9	1.67 ± 0.07	0.08 ± 0.01	2.00 ± 0.18	I	IR	IR	
Prumirim	UB	20 ± 0.0	1.14 ± 0.06	0.19 ± 0.03	1.32 ± 0.01	R	R	R	
Sahy	SS	20 ± 5.0	1.63 ± 0.20	0.08 ± 0.03	1.65 ± 0.37	I	I	R	
Santa Rita	UB	13 ± 2.9	1.80 ± 0.06	0.08 ± 0.01	2.52 ± 0.16	I	I	I	
Santiago	SS	22 ± 2.9	1.66 ± 0.18	0.06 ± 0.02	1.55 ± 0.12	IR	R	IR	
Tabatinga	CA	25 ± 0.0	1.74 ± 0.07	0.09 ± 0.01	2.43 ± 0.37	I	I	I	
Toque-Toque	SS	20 ± 5.0	1.46 ± 0.17	0.11 ± 0.04	1.54 ± 0.05	R	R	IR	
Ubatumirim	UB	65 ± 25.0	2.41 ± 0.25	0.03 ± 0.01	2.98 ± 0.19	D	D	D	
Una	SS	23 ± 5.8	1.47 ± 0.35	0.10 ± 0.04	1.51 ± 0.56	I	R	R	

Figure 2 Examples of the remotely measured beach characteristics.

(A) Fazenda, a beach with dissipative features and (B) Martim de Sá, a beach with reflective features. “Blue lines” represent intertidal width, “red lines” represent the surf-zone width, and the yellow arrows represent the number of waves in the surf zone. The intertidal height was calculated using the “elevation profile” function provided in Google Earth Software. Parameters for establishing exposure degree are shown in Fig. S1. Maps data: (A) ©2021 Google, CNES/Airbus, Maxar Technologies; (B) ©2021 Google, Maxar Technologies.

We used two methods to assess the value of each characteristic. As performed by Harris, Nel & Schoeman (2011), we first measured environmental characteristics from a single image taken on the closest day available to the date of the in-situ beach sampling. However, beach characteristics, such as tide and surf-zone dynamics, are susceptible to daily variability brought by climatic and oceanography variables, such as wind direction and intensity, and storms-driven changes in the surf-zone climate (Ortega et al., 2013; Roberts, Wang & Puleo, 2013). For this reason, we also estimated the value of beach characteristics based on the mean value obtained from five satellite images, using the images closest to the day of the in-situ sampling. This dual approach of estimating beach characteristics allows us to compare if temporal variations can affect the accuracy of the method and also evaluate which metric is most susceptible to this temporal variability. We avoided images that were visibly taken during high tide, as an estimation of intertidal parameters would be misleading in such instances, and used images where the intertidal area was clearly exposed. Additionally, images where low quality or conditions (e.g., very pixelated, taken during cloudy conditions) hindered a proper measurement of beach characteristics were not included in the assessment.

This procedure was carried out for all 87 sites sampled in the 29 target beaches, and this data was used in further analyses to build the predictive model (train dataset, Fig. 3). To test the predictive model, we also processed satellite images for 30 additional beaches from the study area (n = 77 beach sites, as some beaches were small and were considered as a single unit—beaches with less than 100 m long—or divided into only two sites –beaches with less than 300 m long) (test dataset, Fig. 3). The morphodynamic type of those beaches was obtained from literature reports (e.g., Amaral & Denadai, 2011; Souza, 2012; Checon et al., 2018), with the support from the slope and sediment grain size data available from a partner project (see resulting classification on Table S2). This train dataset was used to test the robustness of the model developed with the training dataset in predicting beach type. Finally, to extrapolate the results from the predictive model to the whole North Coast of São Paulo and derive an overall characterization of beach morphodynamics, we carried out the satellite image processing for the remaining continental beaches of the study area, which were divided into sites according to the beach length (n = 73 beaches, 144 sites). We only excluded severely squeezed beaches, which are present in some areas, as the intertidal cannot be properly estimated. This third dataset (extrapolation dataset, Fig. 3) was used only to predict the beach types based on the results from the model, provided the model was robust enough to support this classification.

Figure 3 Summary of the methods for the development, validation and extrapolation of predictive model for morphodynamic type classification using conditional inference tree, following Harris, Nel & Schoeman (2011).

a beaches with previously known morphodynamic classification;b beaches of the study area not included in the train and test dataset; c internal validation was made using a random subset of sites from the train dataset (n = 25), whereas external validation was made using sites from the test dataset.

Data analyses

Environmental and remote characterization

Our first approach was to test whether the point-based (i.e., single images) or the mean-based (i.e., five images) of intertidal width and height, as those had counterparts measured in the field, were best correlated to the field-based measurements using a Pearson correlation (r). To test the null hypothesis of equal correlations between methods for each variable, we used the confidence intervals (CI) methods described in Zou (2007). If the estimated CI for the differences in correlation between the two methods does not reach 0, then the null hypothesis is rejected, and correlation coefficients are considered different. The method with the highest correlation with the field counterparts was used in further analyses.

To visualize the distribution of 87 sampled beach sites along the morphodynamic continuum, we employed a Principal Component Analysis (PCA). The environmental matrix used to characterize the sites was compiled by the variables obtained by in-situ measurements: mean grain diameter, intertidal width, beach slope and the calculated Beach Index. Due to the different scales of the unit measurements of each variable, the matrix was normalized before the analysis.

Model building and validation

To develop the classification scheme for beach morphodynamic type, a conditional inference tree analysis (CTREE) was carried out, the same method used in Harris, Nel & Schoeman (2011). CTREE is part of the tree-based methods, which are useful to build a predictive model to classify outcomes based on the variation of predictive variables (Hothorn, Hornik & Zeileis, 2006). The CTREE uses an unbiased recursive partitioning algorithm which builds the predictive model in two steps: using a statistical inference based on permutation p-values to select predictor variables, then defining the split point value for the variable to determine a specific category/response, which reduces the bias in variable selection (Hothorn, Hornik & Zeileis, 2006).

The accuracy of the classification model was tested using two methods. First, we tested the accuracy by randomly separating the initial dataset in a train (70% of sites, n = 62) and a test dataset (30% of sites, n = 25). The model was built with the training dataset, and the classification scheme power was tested by predicting the beach type with the test dataset and comparing it with the in-situ classified types (internal validation, Fig. 3). For the second method, to evaluate the obtained classification scheme, the additional 77 beach sites with previously known beach type classification were then classified using the CTREE scheme. The results of the beach type predicted by the classification scheme was contrasted with the known morphodynamic type to test the accuracy of the model (external validation, Fig. 3). As accuracy is best tested by multiple metrics (Cutler et al., 2007), we used four metrics to assess the robustness of the model: the sensitivity (i.e., probability of correctly estimating the right elements of a given category; true positive) and specificity (i.e., probability of correctly estimate the elements that do not belong in a given category, true negatives), the percentage of correct classification (PCC), and the Cohen’s weighted kappa (k). The PCC is calculated as the percentage of the predicted classification that agrees with the field-based classification. The weighted kappa estimates the agreement between the qualitative classification between the predicted and field-based classification, but it also takes into account the likelihood of the agreements occurring by chance and the magnitude of disagreements (i.e., whether disagreement was between similar or more distinct classes) (Cohen, 1968). Higher k values indicate stronger associations between the qualitative responses. Confusion matrices (i.e., false positives and negatives in the predicted values for each category) were built to illustrate the predictive power of the model and estimate the mean sensitivity and specificity of the classification, averaged from every response category.

To test whether the model developed on the South African coast could be applied in the conditions bore by the Northern Coast of the São Paulo state, we tested the accuracy of the classification scheme developed by Harris, Nel & Schoeman (2011) in predicting the 77 local sites with previously known classification in the region. The metrics used to assess accuracy were the same as used to predict the robustness of the locally developed classification. It is important to notice that this test is not done to evaluate the robustness of the previous classification, as it is already proven to correctly predict the morphodynamic type of South African beaches with an accuracy higher than 90% (Harris, Nel & Schoeman, 2011). Instead, our goal was to test whether the original model can be robust to identify beach types regardless of geographic region or, as suggested by the authors, if a locally calibrated classification would be needed to improve the method.

Finally, we applied the classification scheme developed to the extrapolation dataset, to obtain complete characterization of the beaches at study area was obtained from this procedure and presented here as a case study. All analyses were carried out using the R Software 4.1.0 (R Core Team, 2021), using the packages party (Hothorn & Zeileis, 2015), caret (Kuhn, 2021), rpart (Therneau & Atkinson, 2019) and factoextra (Kassambara & Mundt, 2020).

Results

Environmental and remote characterization

The satellite-based measurements of intertidal width were strongly correlated to the in-situ measurements, for both point (r = 0.876) and mean values (r = 0.931), indicating the suitability of this approach as a proxy of variables related with width. The correlation, however, was stronger for mean-based values (CI ≠ 0, range = −0.098; −0.031), which were therefore applied in further analyses. Conversely, satellite-based intertidal heights were very poor proxies of field measurements, as they generally overestimated the height of wide and flat profiled beaches, and underestimated the slope of short, steep profiled beaches. For this reason, correlation coefficients were low and negative (point: r = −0.204, mean: r = −0.159), regardless of the method (CI = 0, range = −0.124; 0.033). Due to this very low correlation with in-situ values, we dropped the intertidal height from further analyses.

The environmental characterization showed that sampled sites comprehended a large portion of the recognized morphodynamic gradient of beaches, with beaches ranging from fine to coarse sands, from gentle to steep slopes (Table 1). From the 87 beach sites, 18 were classified as dissipative, 17 as dissipative-intermediate, 22 as intermediate, 11 as reflective-intermediate and 19 as reflective beaches (Table 1). Although for most beaches the same morphological beach type was classified among its sites, a third of the beaches had variations, going from intermediate to dissipative (e.g., Lagoinha, Cidade) and from intermediate to reflective (e.g., Una, Sahy) (Fig. S2). The principal component analysis showed that the sites followed a clear distribution along the morphodynamic gradient evidenced by the first axis (PC 1 = 84.07%), characterized by a gradient of variation in the proxies of beach type (e.g., mean sediment grain size, beach slope, intertidal width) (Fig. 4).

Figure 4 Principal component analysis showing the distribution of beach sites, coded by beach type, along the environmental gradient.

Intertidal width is measured in meters. Slope is measured as tg β, and increasing values indicate steeper profiles. Mean grain size is measured in phi scale, thus higher values indicate lower mean grain size. BI values are dimensionless and higher values indicate more dissipative conditions. Beach Type is coded as D: Dissipative, ID: Intermediate Dissipative; I: Intermediate; IR: Intermediate Reflective; R: Reflective.

Model building and validation

The conditional inference tree model built with the 87 sites dataset showed that intertidal width and degree of exposure were the only variables selected to predict beach type (Fig. 5). Four terminal nodes were identified: Node 7 was characterized by beaches with intertidal areas >57.63 m, which classified dissipative beaches with a small error (5.0%). Node 6 was characterized by beaches with intertidal areas between 57.63 and 34.85 m, which classified most intermediate dissipative beaches (error = 12.5%). Node 4 was characterized by beaches with intertidal areas <34.85 m and having exposure ≤ 2 (i.e., moderately exposed, sheltered and very sheltered conditions) and enclosed most of the intermediate beaches (error = 10%). Node 5 was characterized by beaches with intertidal areas <34.85 m, but having exposure >2 (i.e., exposed and very exposed beaches). This node was composed mostly of reflective beaches; however, there was a higher error rate (38.8%), since all intermediate reflective beaches were also classified within this node. No reflective or intermediate reflective beaches were classified outside this node. For this reason, we grouped the two beach types within this node, which reduced the error to 7.5% (Fig. 5). The final classification was as follows:

Figure 5 Results from the conditional inference trees models show the significant groupings of beaches, according to variations in the remotely measured values of intertidal width and exposure degree.

Beaches are coded as 1–Dissipative, 2–Intermediate Dissipative, 3–Intermediate, 4 –Intermediate Reflective, 5–Reflective. Node 4 represent intermediate beaches (error = 10.0%), Node 6 represent intermediate dissipative beaches (error = 12.5%) and Node 7 represent dissipative beaches (error = 5.0%). Node 5 represents both intermediate reflective and reflective beaches, with a higher error (38.8%), reduced by 7.5% when considering these types as a single unit. Exposure ≤ 2 = very sheltered, sheltered and moderately exposed beaches. Exposure > 2 = exposed and very exposed beaches.

• Dissipative beaches: Intertidal Width >57.63 m (error = 5.0%);

• Intermediate Dissipative beaches: 34.85 m <Intertidal Width <57.63 m (error = 12.5%);

• Intermediate beaches: Intertidal Width <34.85 m; Exposure ≤ 2 (error = 10.0%);

• Intermediate Reflective/Reflective beaches: Intertidal Width <34.85, Exposure >2 (error = 7.5%).

The estimation of the accuracy of this classification through internal validation provided good results. When the classification and prediction datasets were taken as subsets from the initial dataset, the percentage of correct classification (in comparison with the classification based on field measurements) was 92.8%, the weighted kappa indicated a very good agreement (k = 0.897) (Table 2). The confusion matrix shows that only two sites were misclassified using this method (i.e., intermediate beaches classified as intermediate-dissipative) (Table 3). Mean sensitivity and specificity were both high, 92.8 and 97.5% respectively (Table 3). Sensitivity was 100% for all categories, except the intermediate type (70%). Specificity was 100% for all categories, except for intermediate-dissipative (90%).

Table 2 Metrics of the accuracy between the predicted and observed beach type classification.

For test methods, we applied the use of a random subset of the training dataset (n = 25), used only for internal cross-validation of the model, and the use of external (test) sites (n = 77) with a previously known morphodynamic classification, used for validation with the locally developed model and comparing with the classification derived for the South African coast by Harris, Nel & Schoeman (2011), to measure the advantages of a locally calibrated model. PCC: Percentage of correct classification, Kappa: Weighted Cohen’s Kappa. For PCC and Kappa, values closer to 1 indicate perfect agreement.

Test method	Classification	
Random sites (n = 25)	Local	Previous*	
Sensitivity	0.928	–	
Specificity	0.975	–	
PCC	0.920	–	
Kappa	0.897	–	
Test Sites (n = 77)			
Sensitivity	0.950	0.348	
Specificity	0.981	0.746	
PCC	0.948	0.312	
Kappa	0.907	0.019	
Notes.

* Based on Harris, Nel & Schoeman (2011) classification scheme.

Table 3 Confusion matrix for the two methods for testing the accuracy of the predictive model (random sites, test sites).

Diagonal values (shaded area) show the positive predictions (predicted = observed), whereas non-diagonal values show the negative predictions (predictions ≠ observed) for each beach morphodynamic type.

Observed beach type	Predicted beach type	
	Local	Previous*	
	Dis	ID	Int	IR/R	Dis	ID	Int	IR/R	
Random Sites (n = 25)									
Dissipative (Dis)	5	0	0	0	–	–	–	–	
Intermediate-Dissipative (ID)	0	5	0	0	–	–	–	–	
Intermediate (Int)	0	2	5	0	–	–	–	–	
Intermediate-Reflective/Reflective (IR/R)	0	0	0	8	–	–	–	–	
									
Test Sites (n = 77)									
Dissipative (Dis)	6	0	0	0	2	4	0	0	
Intermediate-Dissipative (ID)	1	16	2	1	0	7	13	0	
Intermediate (Int)	0	0	30	0	0	0	1	29	
Intermediate-Reflective/Reflective (IR/R)	0	0	0	21	0	0	7	14	
Notes.

* Based on Harris, Nel & Schoeman (2011) classification scheme.

When testing the accuracy of the classification scheme to predict the beach type of the 77 sites of the test dataset for external validation, the percentage of correct classification was even higher, reaching 94.8% of agreement. The weighted kappa indicated a very good agreement between predicted and observed morphodynamic types (k = 0.907) (Table 2). In the confusion matrix, five of the 77 sites were misclassified and these mismatches were evenly distributed among the categories, with most occurring among close morphodynamic types, except for a single intermediate-reflective/reflective site which was predicted to be intermediate dissipative (Table 3). Mean sensitivity and specificity were also high, 95 and 98.1% respectively (Table 2). Sensitivity was 100% for all categories, except the intermediate-dissipative (80%). Specificity was 100% for all categories, except for intermediate-reflective/reflective (98%).

Finally, when predicting the beach type based on the previous classification scheme developed for the South African coast, the accuracy was very low. Only a third of the sites were correctly classified (PCC = 34.8%), and the weighted kappa indicates a poor agreement (k = 0.019) (Table 2). The confusion matrix shows that 53 of the 77 sites were misclassified by the predictive model (Table 3). Sensitivity was higher for predicting the intermediate-reflective/reflective type (67.7%) but was low for dissipative (33.3%) and intermediate-dissipative types (35.0%), and very low for the intermediate type (3.3%), where only a single site was correctly classified. For the latter category, most sites were erroneously classified as intermediate-reflective/reflective type (Table 3). Specific values were higher than sensitivity, being 100% for dissipative and 92.9% for intermediate-dissipative but having lower values for intermediate (57.4%) and intermediate-reflective/reflective sites (48.2%). Mean sensitivity and specificity values were 34.8% and 74.6%, respectively (Table 2).

Model extrapolation

The extrapolation to the whole study area allowed us to provide a characterization of the 310 beach sites along 147 beaches of the study area (Fig. 6). Considering the 4 beach categories delimited by the model, most sites were classified as intermediate (n = 134, 43.5%). Although the second most common type was intermediate reflective/reflective (n = 76, 24.6%), which includes two beach types due to the inability of the model to differentiate among them. The intermediate dissipative and dissipative types were assigned to 53 (17.2%) and 45 (14.6%) sites, respectively. There is a clear distinction in beach type distribution along the study area. At the southernmost portion the coast is more exposed, and although all beach types are found, this region is where intermediate reflective/reflective beaches sites are most predominant, while more dissipative types are found towards the São Sebastião Channel (Fig. 6A). At the northern area of the São Sebastião municipality, beaches are heavily sheltered by the São Sebastião Island, and most sites are classified as intermediate (Fig. 6B). The central part of the study area is also divided in two; the southern is heavily sheltered inside the Caraguatatuba Bay, and most beaches are of intermediate dissipative and dissipative types, and the northern area is a more exposed shore, with most beaches belonging to intermediate and intermediate reflective/reflective type (Fig. 6B). The coast of the municipality of Ubatuba is a very heterogeneous coast, with the highest number of beaches, due to the heavily jagged coast, especially in the southern area (Fig. 6C). This result in a mosaic of beach sites of distinct types very close to each other, with sheltered beach sites within embayments, ranging from intermediate to dissipative type, and exposed beach sites on the outside of these embayments, with beaches ranging from intermediate to intermediate reflective/reflective type. At the northernmost portion of the study area, the coast is less jagged, and beaches are more exposed, with most sites classified either as intermediate reflective/reflective and dissipative types (Fig. 6C).

Figure 6 Classification of beach type along the study area (∼200 km of coastline) based on the developed model.

(A) Southern region (south coast of the municipality of São Sebastião to the São Sebastião Channel); (B) Central region (north coast of the municipality of São Sebastião and municipality of Caraguatatuba); and (C) North region (municipality of Ubatuba).

Discussion

Our results corroborate the effectiveness of the method developed by Harris, Nel & Schoeman (2011) and reinforce the use of satellite imagery to map and classify beach morphodynamic types. However, the discrepancy between our classification scheme and the original classification developed for the South African coast (Harris, Nel & Schoeman, 2011) highlights the need for local calibration of the method for an improved remote assessment of beach morphodynamics. This necessity was already suggested by Harris, Nel & Schoeman (2011) but, to our knowledge, this is the first study to empirically test it. Although developing a local classification requires more effort (i.e., calibrating the model according to the local environmental features), it can be easily achieved when the beach type of some of the local beaches is already known. The other necessary step to achieve a regional classification of sandy beaches, i.e., the measurements on satellite images, can be easily done, especially considering the use of accessible software, such as Google Earth.

The accuracy of the method in classifying beach type relies on the efficiency of remote-sensing metrics in representing in-situ measurements. Here, we show that the remote measurements of intertidal width were very strongly correlated to in-situ observed values. We also found that point-based measurements reflected width measurement accurately; however, the use of a temporal series of images provided a better correlation than the use of a single image, which reflects the natural temporal variability of physical characteristics of sandy beaches (Wright & Short, 1984; Masselink & Pattiaratchi, 2001).

On the other hand, measuring the height of the intertidal using satellite images was a very inaccurate approach, tending to severely overestimate the inclination of beaches with wide intertidal areas, which are commonly less slopy, resulting in a negative correlation between field and satellite measurements. It is important to notice, however, that this inaccuracy may be overcome with the use more sophisticated image processing tools that can provide a higher level of detail to assess the slope of the beach profile (Splinter, Harley & Turner, 2018), if the application requires a more profound focus on this aspect. Regardless, the method by Harris, Nel & Schoeman (2011) proves robust enough even if the software was unable to estimate the inclination of local beaches.

Importantly, our results corroborate that this method has a >90% predictive power, as found by Harris, Nel & Schoeman (2011). Similarly, their model was also unable to separate the intermediate-reflective from the reflective beaches, which were grouped into a single node, as was done here based on the results of the model. This decision is supported by the fact that all intermediate-reflective and reflective sites were within the same node; if more than one node included beaches of these morphodynamic types, then grouping within a single node would be much less justified. Remote methods involving processing of aerial images can be an alternative or complementary method to support distinction of beaches at the reflective end of the continuum (Browne et al., 2006).

In comparison to the Harris, Nel & Schoeman (2011) classification for South African beaches, both models included intertidal width as a predictor, which is expected given that this variable is associated with variations in morphodynamics and applied in indices to classify beach states, such as the Beach Index (BI) and Beach Deposit Index (BDI) (Wright & Short, 1984; McLachlan & Dorvlo, 2005; Defeo & McLachlan, 2013). However, while intertidal width was the sole predictor in Harris, Nel & Schoeman (2011) model, ours included the importance of exposure for classifying beaches remotely. The South African coast, especially in the west and eastern shore, tends to be highly exposed, with few barrier islands and sheltered beaches (McLachlan, Wooldridge & Dye, 1981). When we applied the model developed by Harris, Nel & Schoeman (2011), where the exposure degree is not included, the classification tended to consider beaches as more reflective than they actually are, and almost all intermediate beaches would be classified as intermediate-reflective/reflective type. At the North coast of São Paulo, however, beaches tend to experience a varying degree of exposure, especially at the northern portion, where the coastal mountain range reaches into the sea at several points, resulting in a heavily jagged coastline, with the presence of many nearshore islands that act as a barrier to direct wave exposure (Souza, 2012). This panorama results in many beaches, especially of intermediate type with moderate to narrow intertidal zones, being located inside bays and sheltered from wave action, which explains the importance of exposure in distinguishing these beaches from the also narrow reflective and intermediate reflective beaches. The relevance of this local feature also highlights the need to calibrate the classification scheme using local conditions.

Our study was developed on a smaller geographical region than that used by Harris, Nel & Schoeman (2011); however, it included a higher number of sites for local validation. Using 87 sites for model development, and another 77 for model validation, we obtained estimates of beach type with a confidence level higher than 90%. Nevertheless, a shared shortcoming of this study and the previous one is the lack of validation for regions under meso- and macrotidal regimes. In these areas, the width of the beach is bound to be higher than those of microtidal regimes, and cut-off values are likely to be very distinct (Harris, Nel & Schoeman, 2011). Furthermore, as the method relies on the use of images taken during a limited range of period, it does not capture the dynamic nature of beach environments. Here, we used mean values from multiple images to reduce the bias of static measurements; however, we acknowledge the method does not allow to observe short-term changes in local dynamics, such as those induced by storms, which can be captured by local continuous monitoring (Splinter, Harley & Turner, 2018) or by other remote sensing methods, such as argus video monitoring (Kroon et al., 2007) or satellite imagery (Pérez Valentín & Müller, 2020). Some other shortcomings are the applicability only in images taken during cloudless conditions, associated with fair weather, a variable that affect local hydrodynamics, and the limitation in assessing metrics, especially intertidal width, in severely squeezed beaches or those with darker sands (i.e., monazite sands). Also, when the goal is to identify particular hydrodynamics (e.g., currents, sediment transport) and morphodynamic features (e.g., nearshore bars), then methods based on image processing are more suitable alternatives (Deronde et al., 2008; Román-Rivera & Ellis, 2019), and may support a more refined classification of beaches within the intermediate range (Browne et al., 2006).

Despite a few intrinsic shortcomings, the method by Harris, Nel & Schoeman (2011) provides advantages. For instance, it provides a fast, cheap, and accessible solution to classify general patterns of beach type, without the need for processing of remote images, with accuracy and at a morphodynamic resolution relevant for ecological studies and management (Harris, Nel & Schoeman, 2011; McLachlan, Defeo & Short, 2018). Its accessibility allows for a large-scale application and mapping of beach ecosystems, such as the one made of the study area. Finally, even though it is based on the analysis of snapshot pictures, it could be used to capture temporal changes in beach state if applied on a temporal basis, although this pends further evaluation.

Implications for management

The method applied here was very accurate in detecting beach morphodynamic type, which can be a valuable tool to characterize coastal areas with many beaches with remote or difficult access, as well as reduce costs with field samplings to assess and monitor relevant beach characteristics. Furthermore, identifying the morphodynamic type of beaches is an important aspect for management purposes, as this metric is highly related to biodiversity patterns (Defeo & McLachlan, 2005), to the response and recovery from impacts, and also have direct repercussions on the quality and quantity of ecosystem services (McLachlan et al., 2013; Harris et al., 2014; McLachlan & Defeo, 2017).

For instance, the impact of organic and inorganic pollution, a recurrent problem in coastal areas worldwide, is known to be stronger in dissipative beaches, especially in sheltered conditions, due to the slower recovery linked to the lower permeability of the beach face (De La Huz et al., 2005; Harris et al., 2015). In this regard, the knowledge of beach type coupled with information of potential impacts, such as the risk of oil spills and/or presence of sewage outfalls in the beach, common problems in our study area (Zanardi-Lamardo, Bícego & Weber, 2013; Santos & Turra, 2017), can significantly help managers to predict and minimize impacts on beaches and their biodiversity. Plastic pollution, one of the most pressing problems in the marine environment, representing up to 80% of marine waste (Auta, Emenike & Fauziah, 2017), is also suggested by recent studies to have its accumulation linked to the energy and morophodynamic of beaches (Tsukada et al., 2021; Wilson et al., 2021). Thus, there is great potential for integrating morphodynamic types and the dynamics of litter on beaches to support management strategies (Fanini et al., 2021).

Aside from classifying beach morphodynamics, the measurements of beach features can be a rapid way of assessing information relevant for other management and economic purposes. For instance, the intertidal width, a metric with a strong correlation between local and satellite-based measurements, can be used to infer the suitability of beaches for tourism activities. Studies assessing tourist preferences found that beaches with large intertidal areas are usually preferred by general tourists, likely due to the available space, as well as the smooth slope commonly associated with wide beaches, which facilitates the practice of recreational activities (Onofri & Nunes, 2013). In contrast, surfers prefer beaches with a higher number of waves and wider surf-zones (Philips & House, 2009), metrics that were easily assessed in the present study.

Additionally, beaches with restricted supralittoral areas (i.e., with the intertidal located too close to the upper beach limits), especially those with armoring structures, may be more susceptible to climate change impacts, due to being more prone to rises in the mean sea level (Pontee, 2013; Leo et al., 2019). Thus, if beach width metrics are coupled with information of anthropic structures in beaches, then susceptibility to climate change impacts could be inferred. Remote imaging has also the potential to estimate features characteristic of urbanization such as shoreline occupation, suppression of vegetation, and presence of coastal armoring structures (Hall & Hossain, 2020; Morgan et al., 2022), which could be incorporated into an integrated framework with morphodynamic classification to assess this vulnerability.

Finally, the corroboration of the correlation between remotely measured metrics with in-situ characteristics represents an opportunity to use this tool to support predictive models of biodiversity, such as habitat suitability modeling. The use of remote sensing to aid niche modeling to predict and map biodiversity is common in terrestrial ecological studies (Goetz et al., 2010; Leitão & Santos, 2019; Wang & Gamon, 2019). However, its use to predict biodiversity in beach ecosystems is very limited (e.g., Marzialetti et al., 2021), and most remote sensing tools in coastal studies are used to assess and monitor physical characteristics (Kroon et al., 2007; Mars & Houseknecht, 2007; Luijendijk et al., 2018). Beach morphodynamic characteristics are considered important predictors of biodiversity, both from benthic assemblages, as well as surf-zone fishes (Defeo & McLachlan, 2005; McLachlan & Dorvlo, 2005; Shah Esmaeili et al., 2021). Evaluation of the degree of exposure may also be used to predict ecological processes, such as nursery potential (Oliveira & Pessanha, 2014) and biological invasions (Hampton & Griffiths, 2017). For this reason, further studies could aim to uncover the potential of the remote sensing approach in contributing to modeling biodiversity and sustainable use in sandy beaches.

Conclusions

In synthesis, our results corroborate the effectiveness of the method developed by Harris, Nel & Schoeman (2011) to map beach morphodynamics based on remote measurements of beach characteristics and reinforce that the use of satellite imagery may be an important tool for the management of sandy beach ecosystems. It also shows the importance of regionally developed classification schemes, as our results were slightly different from the one obtained for the South African coastline, with exposure degree, aside from intertidal width, being important to remotely classify beaches at the study area. The extrapolation of the model allows for a fast assessment of beach morphodynamics types, which can aid monitoring environmental changes and managing tourism preferences and environmental susceptibility to local impacts.

Supplemental Information

Supplemental Information 1 Examples of beaches with different degrees of exposure

(A), (B) exposed and very exposed beaches, respectively, with little to no curvature or embayment caused by prominent headlands, and no sheltering from barrier islands. (C) moderately exposed beach, with sheltering caused by curvature of the headlands, that shelters and dissipates wave energy; and (D) sheltered and very sheltered beaches, located within embayments, with very exposure to wave action. Maps data: (A), (C) and (D) ©2021 Google, Maxar Technologies; (B) ©2021 Google, CNES/Airbus, Maxar Technologies

Click here for additional data file.

Supplemental Information 2 Examples of beaches with varying morphodynamic condition. In both cases, the right corner is more sheltered due to a prominent headland

In (A) (Una beach), the right corner is a moderately exposed intermediate beach type, whereas the left corner present wave exposed reflective conditions. In (B) (Lagoinha beach), the right corner is a sheltered dissipative beach type, whereas the right corner is moderately sheltered intermediate beach. Maps data: (A) ©2021 Google, CNES/Airbus; (B) ©2021 Google, Maxar Technologies

Click here for additional data file.

Supplemental Information 3 Summary of the in-situ characteristics of the 87 sampled sites

Beach type is coded as: D: Dissipative, ID: Intermediate-dissipative; I: Intermediate; IR: Intermediate reflective; R: Reflective.

Click here for additional data file.

Supplemental Information 4 Morphodynamic state of the 77 beach sites used as to validate the classification model

Beach type are coded as: D: Dissipative; ID: Intermediate-dissipative; I: Intermediate; IR: Intermediate Reflective; R: Reflective.

Click here for additional data file.

The authors would like to thank L Barbosa, IA Laurino, MN Raganin, M Rached, BP Cunha, R Rodrigues, T Gomes, CR Bilatto, J Ramalho, I Chechin, G Rabelo, and AS Minato for their help with sampling procedures.

Additional Information and Declarations

Competing Interests

Author Contributions

Data Availability

Guilherme N. Corte is an Academic Editor for PeerJ. The other authors declare that they have no competing interests.

Helio Herminio Checon conceived and designed the experiments, performed the experiments, analyzed the data, prepared figures and/or tables, authored or reviewed drafts of the paper, and approved the final draft.

Yasmina Shah Esmaeili conceived and designed the experiments, performed the experiments, analyzed the data, prepared figures and/or tables, authored or reviewed drafts of the paper, and approved the final draft.

Guilherme N. Corte conceived and designed the experiments, performed the experiments, analyzed the data, authored or reviewed drafts of the paper, and approved the final draft.

Nicole Malinconico analyzed the data, authored or reviewed drafts of the paper, and approved the final draft.

Alexander Turra conceived and designed the experiments, authored or reviewed drafts of the paper, and approved the final draft.

The following information was supplied regarding data availability:

The raw data used for the environmental characterization of beaches of the train dataset and the characteristics of the beaches of the test dataset are available in the Supplementary Tables.

The remaining raw data is available at Zenodo: Checon, Helio Herminio. (2022). Dataset associated with the manuscript “Locally developed models improve the accuracy of remotely assessed metrics as a rapid tool to classify sandy beach morphodynamics” [Data set]. Zenodo. https://doi.org/10.5281/zenodo.6476540.

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
