# Peer review of "Locally developed models improve the accuracy of remotely assessed metrics as a rapid tool to classify sandy beach morphodynamics"

_PeerJ, doi:10.7717/peerj.13413_

## Round 0.1 · original submission · Major Revisions

Two referees have provided comments on your submission. Both see merit in the work and recommend that the article could be accepted after revision, I agree.

Reviewer 1 ·

Basic reporting

I have now read "Locally developed models improve the accuracy of
remote sensing as a rapid tool to assess and classify
sandy beach morphodynamics (#70546)".

i do not think the manuscript currently does enough to contextualize the work. For example, not enough text explains the point of measuring beach state for ecological studies, and what has been done previously to measure beach state (especially via remote sensing).

1) the paper is largely motivated by ecological studies of beach state, but can the paper demonstrate how knowing if "reflective, intermediate, or dissipative" is specifically useful for ecological and biodiversity studies? please include this information in the intro to motivate the paper.

2) the paragraph starting line 41: Defining beach state has been a common activity for coastal geomorphologists since atleast Guza and Inman 1975 and Wright and Short 1984. I think the paper needs to look in the literature (including and beyond these two papers) and explain what beach state is, why its even useful (i.e., understanding hydrodynamic conditions and sediment transport patterns) and why specifically its even useful for ecological studies (i.e., the motivation from the first paragraph). To this end, new text is needed to specifically justify Line 57-59; New text is needed on the specific examples from the cited papers on Line 55-57.

3) The paragraph on line 24 ends by discussing how beach state has been determined using the Harris 2011 method. What other methods have been used to determine beach state from remotely sensed imagery? what are the benefits/drawbacks of the Harris et al 2011 method vs others? The paper should include a literature review on beach state because many coastal geomorphologists have worked on remote sensing of beach state. Wright and Short has been cited over 2000 times, and 500 of those citations have come in the past 4 years, so it would be worth looking through those abstracts to see what else has been done to monitor morphodynamics remotely (aside from Harris et al 2011).

Experimental design

Section starting on 110 and 113 : I don't understand how/if tidal correction was done? For both field and remote sensing, the intertidal width would be strongly dependent on the when in the tidal cycle measurements were taken. Please explain the exact way that tidal stage was controlled.

Validity of the findings

I am concerned that the developed predictor is so strongly dependent on intertidal width (Line 258-259). What are the specific implications of this specific finding? Should we expect this model to be general and transferable? how does it compare to Harris' work? What were the controls in that paper? why are they different/same? How about the other studies that used the Harris predictor? have other beach state prediction papers found the same single control?

Reviewer 2 ·

Basic reporting

the submitted manuscript is based on a replication of a reasonably well accepted methodology (proposed by Haris et al 2011) for beach classification in the southeast region of Brazil
The paper is well written and has an adequate structure with a good presentation of the scientific question, presentation of data and a proper methodological approach.

Experimental design

The experimental design of the work was well performed, using methods of data collection and analysis compatible with the problem raised (see comments below)

Validity of the findings

The results bring important contributions on rapid beach classification methods from orbital image platforms (i.e. google earth), which can become a useful tool for management and coastal conservation

Additional comments

Some important points can be raised so that the paper can be accepted for publication
A central point, in my opinion, is the title: although technically using the term 'remote sensing' to describe the process of measuring morphological and topological parameters of beaches is not incorrect, among the scientific community this term is applied when some level of image processing is employed in which spectral (or textural) information is extracted from the images. Using direct distance measurements from the "Google Earth" platform (in which several satellites are used indiscriminately) does not clearly indicate the use of "remote sensing" methods strictu sensu and may mislead the reader about the type of methodology used.
Another important point is that the paper neglects the dynamic aspect of the nature of beaches (and this is also a recurrent criticism of the work of Haris et al 2011 and others who use satellite images in beach classification). Measurements such as slope, wave frequency and surf-zone type are extremely dependent on dynamic conditions that may reflect very different short and long term climatic/oceanographic conditions (which are reflected in the characteristics of the beach profile), and by taking static measurements of these parameters into account, the classification models can yield significant uncertainty. Furthermore, there are other issues that must be raised as problems in the use of satellite images in beach characterization, for example the fact that image collection tends to be biased towards the synoptic characteristics of the beach profile (after all, satellite images are obtained under cloudless conditions, which are associated with fair weather).
In my opinion, the authors in addition to reproducing the methodology of Haris et al (op cit) and comparing the results (convergences and divergences) might consider addressing these issues and constructing a critical review, suggesting improvements and recommendations for improved accuracy and precision for future models that could take into account the morphodynamic aspects of the classification systems.

---

## Round 0.2 · accepted · Accept

Thank you for revising and enhancing the manuscript.